# Potential Effect of Enzymatic Porcine Placental Hydrolysate (EPPH) to Improve Alcoholic Liver Disease (ALD) by Promoting Lipolysis in the Liver

**DOI:** 10.3390/biology11071012

**Published:** 2022-07-06

**Authors:** Hak Yong Lee, Young Mi Park, Dong Yeop Shin, Kwang Hyun Park, Min Ju Kim, Sun Myung Yoon, Keun Nam Kim, Hye Jeong Yang, Min Jung Kim, Soo-Cheol Choi, In-Ah Lee

**Affiliations:** 1Invivo Co., Ltd., 121, Deahak-ro, Seongbuk-gu, Nonsan 32992, Korea; leeapf@nate.com (H.Y.L.); pym07130@hanmail.net (Y.M.P.); sdy1325@hanmail.net (D.Y.S.); 2Department of Oriental, Nambu University, 23 Chumdanjungang-ro, Gwangsan-gu, Gwangju 62271, Korea; khpark@nambu.ac.kr; 3Ubio, Unimed Bldg. 69, Samjeon-ro, Songpa-gu, Seoul 05567, Korea; alswn0512@unimed.co.kr (M.J.K.); smyoon@unimed.co.kr (S.M.Y.); 4Unimed Pharm Inc., Unimed Bldg. 69, Samjeon-ro, Songpa-gu, Seoul 05567, Korea; knkim@unimed.co.kr; 5Korea Food Research Institute, 245, Nonsaengmyeong-ro, Wansan-gu, Wanju 55365, Korea; yhj@kfri.re.kr (H.J.Y.); kmj@kfri.re.kr (M.J.K.); 6Department of Chemistry, Kunsan National University, 558 Daehak-ro, Miryong-dong, Gunsan 54150, Korea

**Keywords:** enzymatic porcine placental hydrolysate (EPPH), alcoholic liver disease, oxidative stress, inflammatory liver injury, fatty liver

## Abstract

**Simple Summary:**

Enzymatic porcine placental hydrolyzing (EPPH) was extracted using enzymatic hydrolysis to extract placental-derived peptides recently reported as excellent bioactive substances, and EPPH was administered to HepG2 cells and alcohol-derived animal models. As a result, it was confirmed that inflammation and antioxidant activity could be regulated through CYP2E1 inhibition. Through these results, we confirmed that the antioxidant and anti-inflammatory effects mediated by CYP2E1 regulation of EPPH have potential for improvement and protection of liver disease in alcoholic liver disease animal models.

**Abstract:**

Alcoholic liver disease is associated with the production of highly reactive free radicals by ethanol and its metabolites. Free radicals not only induce liver oxidation and damage tissues, but also stimulate an inflammatory response in hepatocytes, leading to severe liver disease. In order to improve alcoholic liver disease, enzymatic porcine placenta hydrolysate was studied by exploring various materials. Enzymatic porcine placenta hydrolysate (EPPH) contains various amino acids, peptides, and proteins, and is used as a useful substance in the body. In this study, changes were confirmed in indicators related to the antioxidant efficacy of EPPH in vitro and in vivo. EPPH inhibits an EtOH-induced decrease in superoxide dismutase and catalase activity through inhibition of free radicals without endogenous cytotoxicity. EPPH has been observed to have a partial effect on common liver function factors such as liver weight, ALT, AST, ALP, and GGT. In addition, EPPH affected changes in fat regulators and inflammatory cytokines in blood biochemical assays. It was confirmed that EPPH was involved in fat metabolism in hepatocytes by regulating PPARα in an alcoholic liver disease animal model. Therefore, EPPH strongly modulates Bcl-2 and BAX involved in apoptosis, thereby exhibiting cytochrome P450 (CYP)-inhibitory effects in alcoholic liver disease cells. As a result, this study confirmed that EPPH is a substance that can help liver health by improving liver disease in an alcoholic liver disease animal model.

## 1. Introduction

Alcoholic liver disease is one of the most common diseases in modern people, and alcohol is known to adversely affect various organs of the human body, such as the brain, heart, pancreas, and liver, especially the mechanism that causes liver disease [1,2,3]. The main metabolism of alcohol occurs in mitochondria, where ethanol is oxidized to acetaldehyde by alcohol dehydrogenase (ADH) and then oxidized to acetic acid by the action of ALDH. This series of oxidation reactions requires a steady supply of NAD by the reoxidation reaction from NADH to NAD, and eventually, the above two reactions proceed simultaneously, oxidizing alcohol through alcohol metabolism through the overall redox state of the cell [4,5,6,7].

However, if ethanol is consumed continuously, oxidation of ADH-ALDH alone is difficult, and alcohol metabolism proceeds with the CYP450 2E1 catalase enzyme [8,9,10,11]. Unlike the normal group, in the ethanol intake group, cytochrome P450 enzymes (especially CYP4502E1) are maintained 5 to 10 times more enhanced than in normal people, and this increase in CYP4502E1 increases TG, blood fatty acid levels, and increases the production of active oxygen, resulting in alcohol-related liver damage [12,13].

In addition, an increase in CYP 2E1-ADH induces an immune response, increasing TNF-α, TGF-β, interleukin (IL-1β), and IL-6 [14,15,16,17]. Cytokines stimulate astrocytes to deposit collagen, cause liver fibrosis, and increase free radicals [6,18]. These free radicals are removed by catalase, superoxide dismutase (SOD), and glutathione peroxidase. When the concentration of free radicals generated during alcohol metabolism exceeds the ability of the antioxidant defense mechanism that treats them, liver damage is induced, and the lower the enzyme level, the more severe the liver damage [14,19,20].

The placenta is an important organ that transports nutrients between the mother and the fetus or supplies important endocrine substances to maintain pregnancy, and according to related studies, it is used in various fields such as chronic diseases and pain diseases, paralysis, osteoporosis, and beauty treatment [21,22]. The main components of the placenta are composed of dozens of amino acids, various active peptides, vitamins, sugars, nucleic acids, minerals, and hundreds of enzymes, in addition to essential amino acids [23]. It is also known to play a role in inducing and activating the differentiation and synthesis of various cells and organs in the body and normalizing various body functions by strengthening immunity [24,25].

In this study, enzymatic porcine placental hydrolysate (EPPH) was extracted using an enzyme hydrolysis method to extract placental-derived peptides recently reported as excellent bioactive substances [26,27], and EPPH was administered to HepG2 cell- and alcohol-derived animal models. As a result, it was confirmed that inflammation and antioxidant activity could be controlled through CYP 2E1 inhibition.

Through these results, we hypothesized that the antioxidant and anti-inflammatory effects mediated by CYP 2E1 regulation are very important in the regulation of ALD mechanisms. Specifically, we evaluated the effect of EPPH in in vitro and in vivo models of ALD tailored to the effect of HPE treatment in the early stages of alcoholic hepatitis and steatosis.

## 2. Materials and Methods

### 2.1. Placenta Banding and Basic Composition Analysis

After the shipment of piglets, the placenta was recovered and received from a farming farm certified by HACCP. In particular, farmers received a safe placenta by conducting periodic virus and harmful microbial quarantine and autonomous quarantine tests as seed farms. The supplied placenta was washed with contaminants or pollutants, and activated carbon was treated to remove impurities such as ethanol and odor, and EPPH in the form of white powder was used for the experiment.

### 2.2. Preparation of Enzymatic Porcine Placenta Hydrolysate (EPPH)

Enzymatic porcine placenta hydrolysate (EPPH) prepared by Ubio (Ubio, Seoul, Korea) was used. The porcine placenta enzymatic hydrolysate was washed several times with NaCl to remove the blood and crush the placenta from which foreign substances were removed. The crushed placenta was subjected to hydrolysis and heating after adding papain, a proteolytic enzyme, to inactivate the protease, followed by filtration and adsorption purification. The purified product was prepared by adsorption purification after leaving/concentrating by adding ethanol. In this experiment, a liquid porcine placenta enzyme hydrolysate was used. The porcine placenta enzyme hydrolysate was quantified/diluted based on the total nitrogen content, and the positive sample silymarin (silymarin, Sigma-Aldrich, St. Louis, MO, USA) was prepared according to the treatment concentration by weight and then used.

### 2.3. Cell Culture and Viability Assay

HepG2 cells were cultured in Dulbecco’s minimum essential media (DMEM; Gibco, Billings, MT, USA) with 10% fetal bovine serum (FBS; Gibco, MT, USA) and 1% penicillin–streptomycin (PS; Gibco, Billings, MT, USA) at 37 °C in a 5% CO_2_ incubator. The cells were subcultured at 70%–80% confluence. Cell viability assays were performed using a WST-1 assay kit (ITSBio, INC., Seoul, Korea) according to the manufacturer’s instructions. Briefly, isolated pancreatic islets (2 × 104 cells/well) were seeded into 96-well plates and incubated at 37 °C for 4 h to allow for cell stabilization. Next, the cells were treated with EPPH and a positive control silymarin and incubated for 24 h in a 5% CO_2_ incubator. After incubation, the visible absorbance at 450 nm of each well was quantified using a microplate reader (Sunrise™; Tecan Group Ltd., Männedorf, Switzerland). Each experiment was performed in triplicate. In all subsequent experiments, silymarin was used as a positive control group.

### 2.4. Effect of EPPH on Antioxidant Activity in HepG2 Cell

HepG2 cell injury was measured by superoxide dismutase (SOD, Cayman Chemical Company, Ann Arbor, MI, USA and catalase (CAT, Bio Vision Co., Waltham, MA, USA) levels using commercially available assay kits. Briefly, HepG2 cells (1 × 106 cells/mL) in a 6-well plate were incubated in the presence or absence of EPPH for 1 h. The cells were then stimulated with 200 M ethanol for 24 h; after incubation, the supernatant of the cell lysate was centrifuged, and SOD and CAT levels were measured using a microplate reader.

### 2.5. Animals and Experimental Design

Male C57BL/6J mice, 5 weeks old, were obtained from Orient Bio (Sungnam, Korea). The use of mice was reviewed and approved by the Invivo Animal Care Committee (IACUC Approval Number: IV-RA-07-2106-16). Mice were housed in a constant condition of temperature (22 ± 2 °C) and humidity (55 ± 2%) room on a 12 h light/dark cycle. After adaptation to the environment, the mice were randomly divided into five groups as follows: normal group, control group, 1.03, 3.08, 9.23 mg/kg of EPPH group, and 200 mg silymarin/kg group. The normal group was fed a Lieber–DeCarli control liquid diet (#710027, Dyets, Bethlehem, PA, USA) and 1.03, 3.08, 9.23 mg/kg of EPPH group, and the 200 mg silymarin/kg group was fed a Lieber–DeCarli liquid ethanol diet (#710260, Dyets, Bethlehem, PA, USA). Ethanol was introduced into the diet by gradually mixing it with distilled water from 0% (*w*/*v*) to 3% (*w*/*v*) over a 2-week period for adaptation, and then at a concentration of 5% (*w*/*v*) for the next 6 weeks. Body weight was measured once a week during the feeding period. At the end of the experiment, mice were sacrificed by isoflurane anesthetization followed by exsanguination. The liver samples and blood were collected and used for further experiments.

### 2.6. Serum Biochemical Parameters

Blood samples were centrifuged at 3000 rpm to separate serum and stored at 4 °C for determining serum biochemistry parameters. The serum levels of aspartate aminotransferase (AST), alanine aminotransferase (ALT), alkaline phosphatase (ALP), gamma-glutamyl transpeptidase (GGT), total cholesterol (TC), triglyceride (TG), high-density lipoproteins (HDL), low-density lipoproteins (LDL), and very-low-density lipoproteins (VLDL) were determined by the Hitachi clinical analyzer 7180 (HITACHI, Tokyo, TYO, Japan). Antioxidant enzyme activity was measured by superoxide dismutase (SOD, Cayman Chemical Company, Ann Arbor, MI, USA) and catalase (CAT, Cayman Chemical Company, Ann Arbor, MI, USA) levels using commercially available assay kits. The levels of interleukin-6 (IL-6) and tumor necrosis factor-α (TNF-α) in serum were measured by ELISA kits according to the manufacturer’s instructions (R&D Systems, Minneapolis, MN, USA).

### 2.7. Western Blot Analysis

Liver tissues were homogenized with ice-cold RIPA buffer containing protease inhibitors (Sigma-Aldrich, MO, USA). Samples were centrifuged at 4 °C, 15,000 rpm for 10 min. The total protein concentration was calculated by the Bradford protein assay kit (Bio-Rad, Hercules, CA, USA). Total protein (10–30 µg per lane) was separated by 8% or 10% SDS-poly-acrylamide gels and transferred to PVDF paper (Whatman, GE Healthcare, Freiburg, Germany). The membranes were blocked with 5% nonfat milk and then incubated with the indicated primary antibody overnight at 4 °C. The blots were incubated with the antibodies against PPAR, CYP 2E1, Bcl-2, BAX, and -actin were purchased from Santa Cruz Biotechnology, Inc. (Santa Cruz, CA, USA). The blot was quantified by enhanced chemiluminescence detection (Bio-Rad, Hercules, CA, USA) with LAS 500 mini (GE Healthcare Bio-Sciences AB, Uppsala, Sweden). Actin is an internal control for ensuring the loading is equal.

### 2.8. Histological Evaluation

After sacrificed animals, the right lobe of the liver was collected for histological evaluation. Hematoxylin and eosin (H&E) staining and Oil Red O staining were performed in histological experiments. For the H&E staining, pieces of the liver were fixed and embedded with paraffin following routine processes. The paraffin sections (5 µm) were stained with the HE staining kit (DAKO, Carpinteria, CA, USA). To perform Oil Red O staining, frozen sections (10 µm) were stained by a commercial kit (Sigma-Aldrich, St. Louis, MO, USA) and examined by light microscopy and image software (Leica, Buffalo Grove, IL, USA). NAS was calculated using the method described by Kleiner et al. [28]. NAS is defined as the sum of the histological scores for steatosis, lobular inflammation, and hepatocyte ballooning. Each score was calculated as displayed in Table 1.

### 2.9. Statistical Analysis

Data were expressed as mean ± standard error (SEM). Statistical analysis was performed using one-way analysis of variance (ANOVA), and post hoc comparisons were carried out using Duncan’s multiple-range test. Differences with *p* < 0.05 were considered to be significant.

## 3. Results

### 3.1. Effect of EPPH on Antioxidant Activity in HepG2 Cell

Typical antioxidative enzymes, superoxide dismutase (Figure 1A,B) and catalase (Figure 1C,D), were markedly rescued by EPPH. EPPH treatment at various concentrations (50, 100, and 200 μg/mL) in HepG2 cells showed an increased synthesis of SOD and CAT. The SOD and CAT values of the untreated HepG2 supernatant were 5.69 and 55.14 nmol/min/mL, respectively. In addition, the SOD and CAT values of HepG2 by ethanol were reduced to 4.12 and 33.90 nmol/min/mL, respectively, and as the treatment concentration of EPPH increased, the synthesis degree of SOD and CAT was increased to 4.13~5.37 and 44.93~50.03, respectively.

### 3.2. Serum Biochemical Parameters

The results reported in Figure 2 show that in the activities of serum AST, ALT, ALP, and GGT in control and experimental animals. The results show that EtOH administration significantly (*p* < 0.05) increased the activities of AST, ALT, ALP, and GGT. Administration of EPPH along with alcohol slightly reversed these liver injury markers towards near normal. EPPH significantly reduced ALT and AST concentration dependently (Figure 2A,B), but efficacy was not confirmed in ALP and GGT indicators (Figure 2C,D). Typically, significant results were confirmed by EPPH dose dependently on TG and VLDL indicators (Figure 3B,E). However, in TC, HDL-C, and LDL-C indicators, it is difficult to confirm the concentration-dependent significant results by EPPH, so this part will be resolved through further research in the future (Figure 3A,C,D). Continuous EtOH consumption would increase membrane permeability of the intestine that enables translocation of the endotoxin of the intestine to the liver. These invaders would be recognized by the immune system and propagate immune responses in the whole body system, then could produce and release more inflammatory cytokines and antioxidative enzymes in the liver (Figure 4). In this study, the TNF-α and IL-6 level in the EtOH group was significantly higher than in the normal group, whereas EPPH downregulated the expression level of TNF-α and IL-6 in a dose-dependent manner (Figure 4A,B). The activities of SOD and CAT in the EPPH group were also significantly lower than in the EtOH group (Figure 4C,D). In the EPPH group, the activities of SOD and CAT were both higher than in the EtOH group and restored back to normal concentrations. EPPH showed dramatic ameliorative effects on EtOH-induced liver cytokine accumulation and downregulated = antioxidative mechanisms.

### 3.3. Western Blot Analysis

PPAR, CYP2E1, Bcl-2, and BAX modulate cell survival, death, and lifespan. Alcohol is a well-known reducer of PPAR and Bcl-2 and inducer of CYP 2E1 and BAX. In Figure 5, PPAR and Bcl-2 protein expression was significantly downregulated in liver tissue of the EtOH group than in the control group. The level of PPAR and Bcl-2 in the EPPH treatment group was markedly higher than in the EtOH group. In contrast, expression of CYP 2E1 and BAX was downregulated by EtOH treatment in liver tissue, but EPPH treatment was rescued in a dose-dependent manner. Based on the results, EPPH modulated the survival and programmed cell-death-related molecule expression in the liver, which was induced by alcohol (Figure 5).

### 3.4. Histological Evaluation

Liver histological assessment was applied for the investigation of hepatic lipid accumulation in mice liver. H&E staining of the liver sections (Figure 6) showed massive steatosis in the EtOH group, which was obviously reduced in the EPPH group to a level similar to that of the control group (Figure 6A). Severe liver damage and hepatocellular necrosis on histopathological liver tissues were observed in the EtOH group (Figure 6B). However, such ameliorative effects were observed in the EPPH group in a dose-dependent manner (Figure 6C–E). Lipid-specific staining was performed by Oil Red O staining (Figure 7). As shown in the Oil Red O stained liver section, the distribution of neutral lipids was improved in the EPPH group in a dose-dependent manner (Figure 7C–E). Based on the results presented, chronic EtOH feeding with EPPH could suppress lipid accumulation in the liver through regulation of lipid metabolism. Quantification of H&E staining and Oil Red O staining was performed by histological scoring (Table 2).

## 4. Discussion

Alcoholic hepatitis (AH) is an acute inflammatory condition that develops in the background of alcohol-related liver disease (ALD). ALD spectrums range from simple steatosis to steatohepatitis, fibrosis, and cirrhosis. The underlying molecular mechanisms of ALD pathogenesis are very complex and have not been fully elucidated. However, a new study has been published that oxidative stress plays a role in mediating the inflammatory response and directly triggering liver damage. Oxidative stress refers to an imbalance in the body’s production and elimination of reactive species (including reactive oxygen species and nitrogenous species) and decreased antioxidant production. For this reason, the regulation of oxidative stress through antioxidant and anti-inflammatory effects is one of the important mechanisms for mitochondrial stress, cell signaling, and epigenetic regulation [29,30,31].

Ethanol, one of the main causes of AH, is known to be metabolized by three main pathways. In this study, we focused on the microsomal ethanol oxidation system (MEOS), which contains the cytochrome P450 2E1 (CYP2E1), an enzyme that requires NADPH, which is induced by chronic alcohol exposure [32,33,34,35]. The increase in CYP2E1 after alcohol intake is due to the stabilization of CYP2E1 rather than de novo synthesis. The MEOS pathway metabolizes ethanol to acetaldehyde by converting NADPH+ and O_2_ to NADP and H_2_O to generate reactive oxygen species (ROS). CYP2E1 plays an important role in lipid peroxidation, protein oxidation, and protein nitrification [29,35,36,37,38].

Ethanol metabolism through CYP2E1 produces acetaldehyde as well as ROS with H_2_O_2_, hydroxyl groups (OH-) and carbon-centered OH-. These ROS can be neutralized by a powerful antioxidant defense system. However, chronic alcoholism interferes with the antioxidant system and induces depletion of mitochondrial glutathione (GSH), which impairs hepatocellular resistance to tumor necrosis factor-alpha (TNF-α) and interleukin-6 (IL-6), resulting in increased apoptotic functionality [39,40].

In this study, the in vitro and in vivo efficacy of enzymatic porcine placental hydrolysate (EPPH) was evaluated by linking the antioxidant and anti-inflammatory regulatory mechanisms to ALD disease in HepG2 cells and mice induced by ethanol and an ethanol supplementation diet (Lieber–DeCarli liquid alcohol diet).

Prior to the experiment, the toxicity of EPPH on HepG2 cells induced by ethanol was set by MTT assay to determine the EPPH concentration in subsequent experiments. In the case of the induced group, in which only 200 μM of EtOH was treated to HepG2 cells, the cell viability was 72.2 ± 1.8%, which was significantly lower than that of the normal group. On the other hand, the experimental group treated with EPPH by concentration showed 76.7 ± 2.6% at 25 μg/mL, 76.2 ± 3.0% at 50 μg/mL, 77.0 ± 3.6% at 100 μg/mL, and 74.3 ± 3.8% at 200 μg/mL. No significant difference was found compared to the ethanol-induced group, but no toxicity was found at a concentration of 200 μg/mL or less. Afterward, the experiment was conducted by setting the EPPH to a concentration of 200 μg/mL or less.

As a result of measuring the antioxidant activity of EPPH in vitro using HepG2 cells, the SOD and CAT contents were increased in a concentration-dependent manner in the experimental group treated with EPPH (100 μg/mL and 200 μg/mL) than in the induction group treated with EtOH alone.

Biochemistry parameters, Western blot, and histological evaluation experiments were conducted in vivo in an animal model in which ALD was induced by ingestion of an alcohol-containing diet.

Serum aminotransferase activity has long been useful as an indicator of ALD. In mice induced by ALD by ingestion of an alcohol diet, hepatocyte damage causes changes in transport function and membrane permeability and eventually releases enzymes from the cells into the blood. Therefore, the amount of blood ALT, AST, ALP, and GGT increased rapidly, and it was confirmed that the content of ALT and AST decreased by EPPH treatment.

In addition, alcohol consumption promotes liver fatty acid synthesis and causes alcoholic fatty liver, increasing the number of lipids released into the bloodstream of the liver, as well as reducing immune function and increasing susceptibility to infection through changes in the immune system. EPPH treatment for fatty liver-derived mice increased and decreased dose-dependent TC, TG, HDL-C, and VLDL levels, resulting in decreased blood liver damage perception and lipid factor changes due to alcohol toxicity, and anti-inflammatory and antioxidant efficacy that reduced TNF-α and IL-6 and increased SOD and CAT.

In the Western blot results, the protein expression of CYP2E1, PPARα, Bcl-2, and BAX in the liver tissue for each experimental group was confirmed. In particular, light oil oxidation of CYP2E1 protein received the most attention as the cause of liver damage, and the generation of reactive oxygen species and reduction of antioxidant mechanisms during this process was estimated to be the main causes of liver damage. As a result, significant results by EPPH treatment were confirmed in the CYP2E1, PPARα, Bcl-2, and BAX Western blot results. In particular, the result of EPPH inhibition of CYP2E1 protein is thought to be a major indicator for the treatment of ALD by inhibiting the alcohol oxidation process in the liver and reducing the generation of reactive oxygen species.

Lastly, in histological evaluation, severe hepatocellular damage and infiltration of inflammatory cells and multinuclear cells were confirmed in the EtOH-administered group, whereas the EPPH-administered group significantly reduced cell damage and restored cell necrosis and inflammatory cell infiltration to normal levels. In addition, as a result of confirming the liver tissue with Oil Red O staining, the control group was stained very red due to the accumulation of intracellular triglycerides, whereas the EPPH-administered experimental group significantly reduced the degree of staining, indicating that the deposition of triglycerides in the tissues by alcohol was suppressed. Based on these results, it seems that EPPH effectively reduced hepatotoxicity by reducing alcohol-induced necrosis and damage to liver cells, alleviating inflammation, and inhibiting fat accumulation.

## 5. Conclusions

Alcohol consumption promotes liver fatty acid synthesis and causes alcoholic fatty liver, increasing the number of lipids released into the bloodstream of the liver, as well as reducing immune function and increasing susceptibility to infection through changes in the immune system. EPPH treatment for fatty liver-derived mice increased and decreased dose-dependent TC, TG, HDL-C, and VLDL levels, resulting in decreased blood liver damage perception and lipid factor changes due to alcohol toxicity and anti-inflammatory and antioxidant efficacy that reduced TNF-α and IL-6 and increased SOD and CAT.

In histological evaluation, severe hepatocellular damage and infiltration of inflammatory cells and multinuclear cells were confirmed in the EtOH-administered group, whereas the EPPH-administered group significantly reduced cell damage and restored cell necrosis and inflammatory cell infiltration to normal levels. Based on these results, it seems that EPPH effectively reduced hepatotoxicity by reducing alcohol-induced necrosis and damage to liver cells, alleviating inflammation, and inhibiting fat accumulation.

## Figures and Tables

**Figure 1 biology-11-01012-f001:**
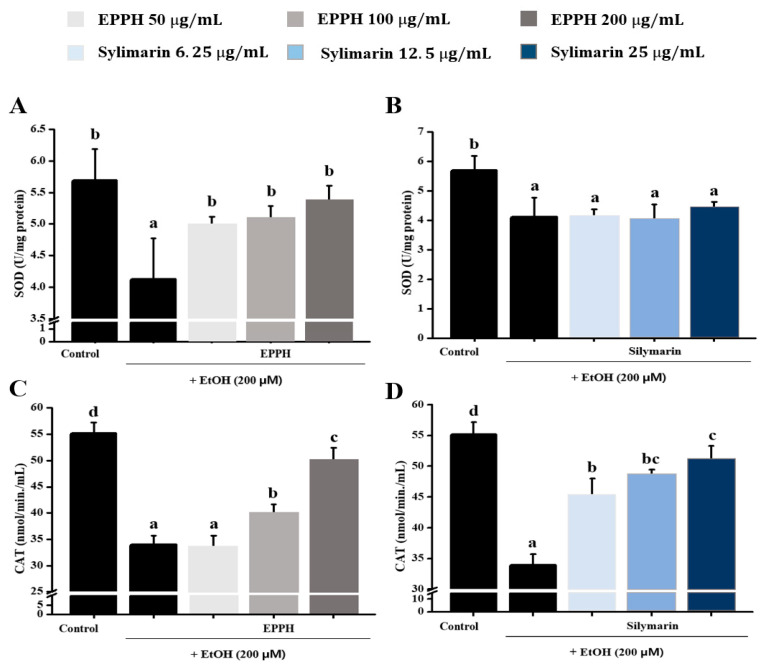
Effect of EPPH extract on EtOH-induced antioxidant enzymes on HepG2 cells. SOD activity with (**A**) EPPH and (**B**) silymarin. CAT activity with (**C**) EPPH and (**D**) silymarin. The data are expressed as the mean ± SD (*n* = 3), and different letters (d > c > b > a) indicate a significant difference at *p* < 0.05, as determined by Duncan’s multiple-range test.

**Figure 2 biology-11-01012-f002:**
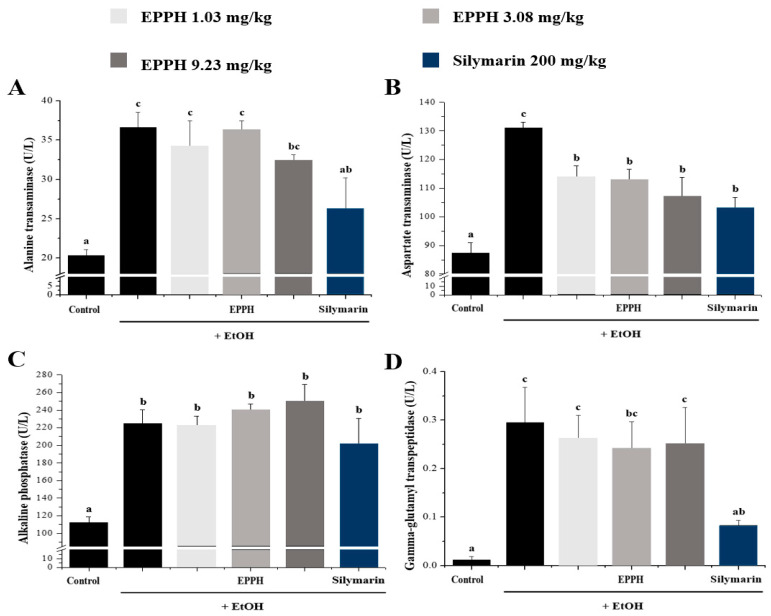
Antioxidant enzymes in blood serum were measured after sacrifice by conventional methods. (**A**) alanine transaminase (ALT), (**B**) aspartate transaminase (AST), (**C**) alkaline phosphatase (ALP), and (**D**) gamma-glutamyl transpeptidase (GGT). Silymarin is a positive control. The data are expressed as the mean ± SD (*n* = 10), and different letters indicate (c > b > a) a significant difference at *p* < 0.05, as determined by Duncan’s multiple-range test.

**Figure 3 biology-11-01012-f003:**
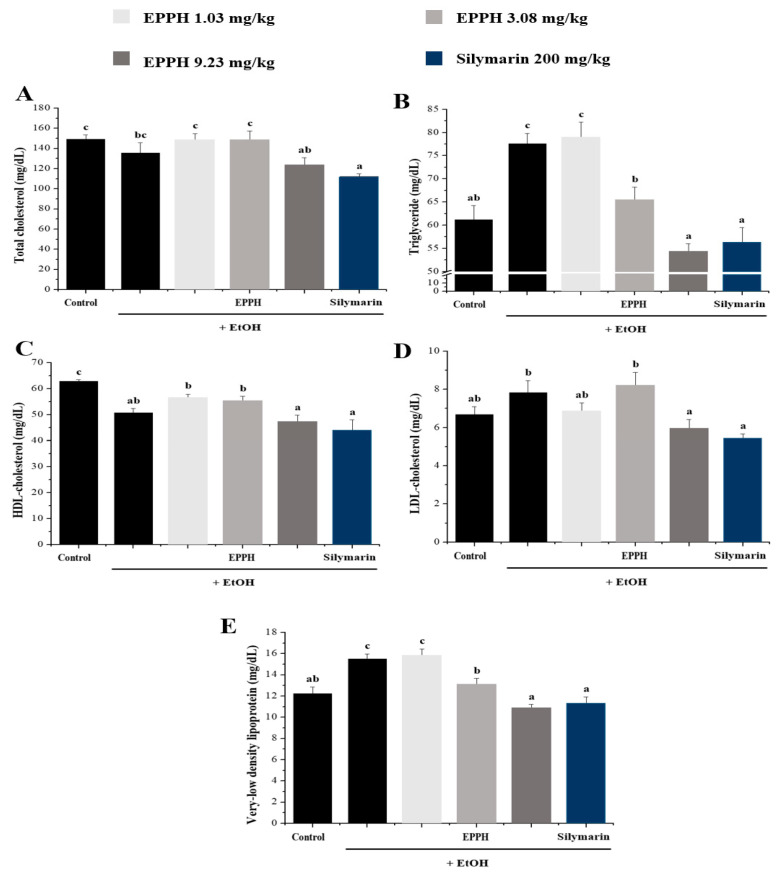
Effects of EPPH on (**A**) total cholesterol (TC), (**B**) triglyceride (TG), (**C**) HDL-cholesterol (HEL-C), and (**D**) LDL-cholesterol (**E**) (LDL-C) mice fed EtOH for 42 days. Silymarin is a positive control. The data are expressed as the mean ± SD (*n* = 10), and different letters (c > b > a) indicate a significant difference at *p* < 0.05, as determined by Duncan’s multiple-range test.

**Figure 4 biology-11-01012-f004:**
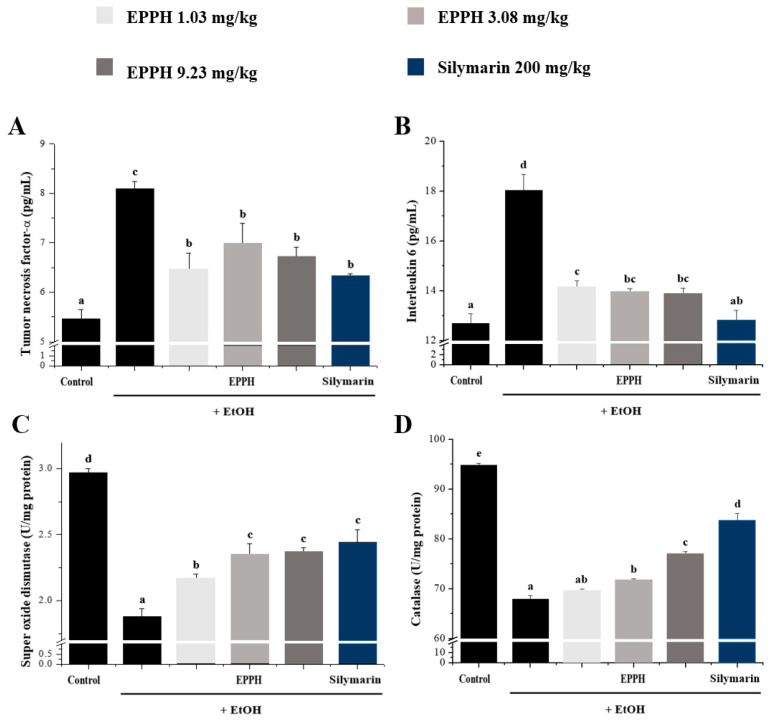
The effect of EPPH on inflammatory markers and lipid metabolism-related markers. Serum levels of (**A**) TNF-α, (**B**) IL-6, (**C**) SOD, and (**D**) CAT in EtOH-induced liver dysfunction in mice. Silymarin is a positive control. The data are expressed as the mean ± SD (*n* = 10), and different letters (d > c > b > a) indicate a significant difference at *p* < 0.05, as determined by Duncan’s multiple-range test.

**Figure 5 biology-11-01012-f005:**
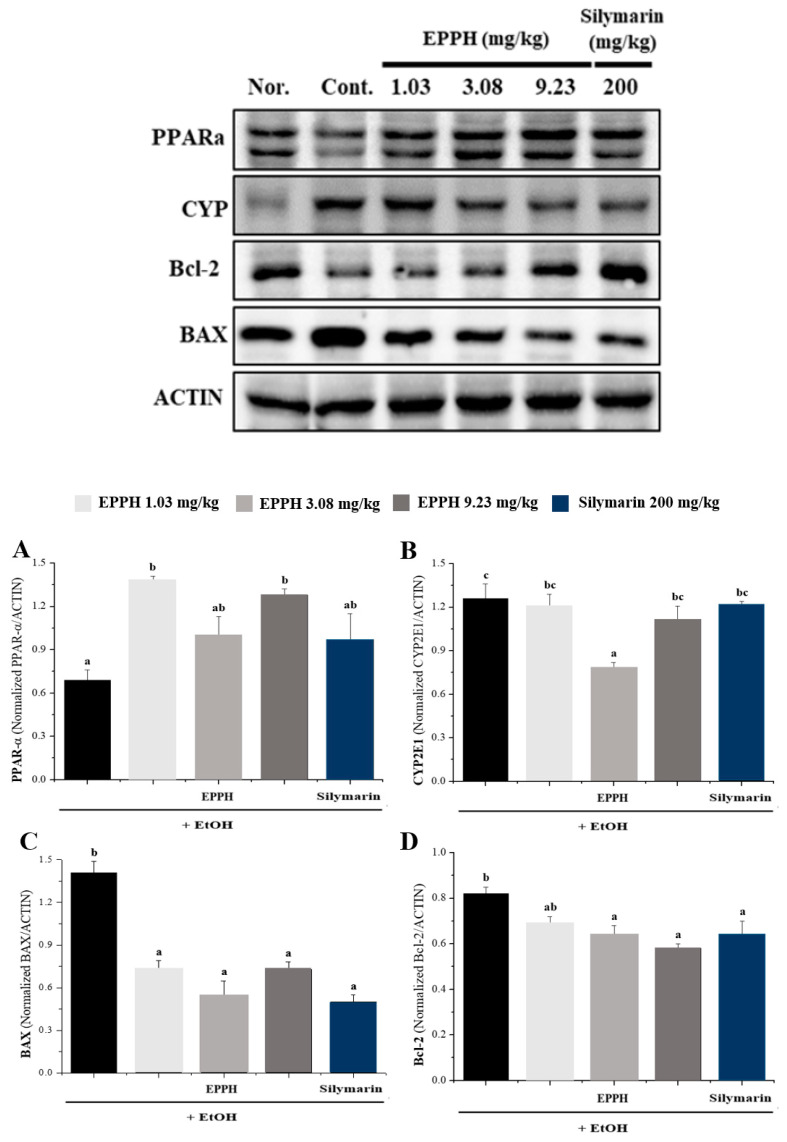
Effects of EPPH on protein expression of PPARα, CYP 2E1, Bcl-2, and BAX in mouse livers, Adult male C57BL/6 mice were administered EtOH without or with EPPH for 42 days, then mouse livers were collected. Protein from mouse livers was analyzed by Western blots for the protein expression of each PPARα, CYP 2E1, Bcl-2, and BAX were analyzed by Western blotting. Silymarin is a positive control. For quantitative comparison of Western blots for each (**A**) PPARα, (**B**) CYP2E1, (**C**) BAX and (**D**) Bcl-2, it was separated and quantified by ACTIN and expressed as a comparative graph. The data are expressed as the mean ± SD (*n* = 10), and different letters indicate (c > b > a) a significant difference at *p* < 0.05, as determined by Duncan’s multiple-range test.

**Figure 6 biology-11-01012-f006:**
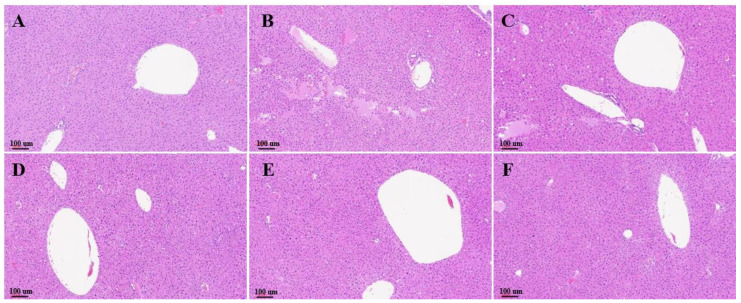
Liver morphometry. EtOH-induced liver damage model mice were orally administered with (**A**) vehicle (saline, normal group), (**B**) control group, (**C**) 1.03, (**D**) 3.08, (**E**) 9.23 mg/kg BW of EPPH, respectively, and (**F**) 200 mg silymarin/kg BW as positive control on a daily basis. After 42 days of treatment, liver morphometry was measured by hematoxylin and eosin staining. Hepatic steatosis is induced in mice exposed to EtOH. Magnification = ×10.

**Figure 7 biology-11-01012-f007:**
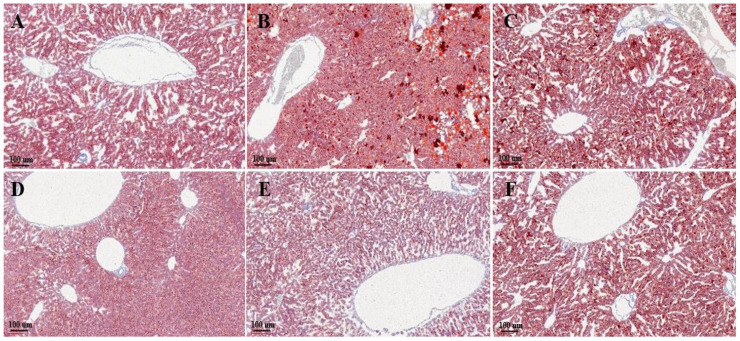
Oil Red O staining of mouse liver tissue treated without or with EPPH. EtOH-induced liver steatosis model mice were orally administered with (**A**) vehicle (saline, normal group), (**B**) control group, (**C**) 1.03, (**D**) 3.08, (**E**) 9.23 mg/kg BW of EPPH, respectively, and (**F**) 200 mg silymarin/kg BW as positive control on a daily basis. After 42 days of treatment, the hepatic steatosis was measured by Oil Red O staining. Magnification = ×10.

**Table 1 biology-11-01012-t001:** Overview of the NAFLD Activity Score (NAS).

Histological Criteria	Severity	Description	Score
Steatosis	Absent	<10%	0
	Mild	10~30%	1
	Marked	31~60%	2
	Servere	>60%	3
Inflammation	None		0
	Moderate	Scattered	1
	Marked	Foci	2
	Severe	Diffuse	3
Hepatocyte Ballooning	None		0
	Few balloon cells		1
	Many cells/prominent ballooning		2

**Table 2 biology-11-01012-t002:** Analysis of histopathology.

(*n* = 10)	NAS Score	Oil-Red-Positive Area (%)
Normal	0.2 ± 0.26 *	5.13 ± 2.53 *
Control	1.8 ± 0.64	11.51 ± 4.35
EPPH (1.03)	1 ± 0.49	13.09 ± 4.22
EPPH (3.08)	0.8 ± 0.46	4.56 ± 2.32 *
EPPH (9.23)	0.2 ± 0.26 *	4.31 ±1.75 *
Silymarin	0.4 ± 0.35	11.41 ± 3.66

* *p* < 0.05 vs. controls. Data are expressed as mean ± SEM. Data analysis was performed by one-way ANOVA and t-test (*n* = 10 mice per group).

## Data Availability

Not applicable.

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
