# Peer review of "Potential Effect of Enzymatic Porcine Placental Hydrolysate (EPPH) to Improve Alcoholic Liver Disease (ALD) by Promoting Lipolysis in the Liver"

_biology, 2022, doi:10.3390/biology11071012_

Round 1

Reviewer 1 Report

This manuscript seeks to determine the protective role of EPPH in the context of liver disease, specifically alcoholic liver disease. Overall, while the manuscript is well organized there are significant issues across all figures and the data is not clearly examined in the results section. The use of letters for statistical comparisons is incredibly difficult to understand and provides more confusion than the text. Please correct and address the following issues:

Line 44: Increase text inside figure

Line 88: collapse to meet previous paragraph.

All figures: It is unclear what these statistics mean. Is each letter is significantly different from control or from a unique group? This needs to be made clearer in the figure legend or methods section. It is incredibly difficult to understand what comparison are being made between treatment groups or interventions.

Figure 7-8: lack appropriate scale bars. These images should be rated and graded by  a pathologist to provide a score of the degree of steatosis.

Author Response

Reviewer 1

  1. Line 44 : Increase text inside figure - We appreciate your comments that may improve the quality of our manuscript. We confirmed the reviewer's request and made changes.

  1. Line 88: collapse to meet previous paragraph – We appreciate your comments that may improve the quality of our manuscript. We confirmed the reviewer's request and made changes.

  1. All figures: It is unclear what these statistics mean. Is each letter is significantly different from control or from a unique group? This needs to be made clearer in the figure legend or methods section. It is incredibly difficult to understand what comparison are being made between treatment groups or interventions.

We appreciate your comments that may improve the quality of our manuscript. We had insufficient interpretation of the statistical results of this manuscript. We used ANOVA for the analysis of the data contained in this manuscript. Because we thought it was necessary to compare all groups to analyze the results.

We tried to check the statistical significance between groups through the ANOVA system for the quantitative values of groups in the statistical part of all experimental results.

The alphabets such as a, b, c and d shown in this manuscript were designated as a, b, c and d from the group with the lowest result regardless of the normal group or the induced group.

If the alpha pets between the groups are the same, it was interpreted as 'it is judged that there is no significance between the groups'. And when the alphabets between the groups are different, it was interpreted as 'it is judged that there is significance between the groups'.

In this experiment, it can be confirmed that the treatment of EPPH in HepG2 cells and experimental animal models showed significant results in various ALD-related indicators based on the control group.

In our revised manuscript, we added a section that lacked interpretation of the data results.

  1. Figure 7-8: lack appropriate scale bars. These images should be rated and graded by  a pathologist to provide a score of the degree of steatosis.

We appreciate your comments that may improve the quality of our manuscript. We modified the part requested by the reviewer and attached it.

Reviewer 2 Report

The authors utilized the EPPH generated from pig placenta to treat HepG2 cells in vitro or alcohol-feeding mice in vivo to assess the anti-oxidant effect in vitro, the effect on liver injury, lipid metabolism, or liver inflammation in vivo.

1. A major concern is HepG2 is not a suitable cell line for investigating the effect of alcohol in vitro, as HepG2 barely expresses ADH or CYP2E1, as shown in a report (Donohue TM, et al. Recombinant Hep G2 cells that express alcohol dehydrogenase and cytochrome P450 2E1 as a model of ethanol-elicited cytotoxicity. Int J Biochem Cell Biol. 2006 Jan;38(1):92-101. doi: 10.1016/j.biocel.2005.07.010. Epub 2005 Sep 6. PMID: 16181800.) Typically, the recombinant HepG2 derivative cells or primary hepatocytes should be used to that end, otherwise, the results based upon parental HepG2 after alcohol treatment cannot be interpreted as the consequences of alcohol treatment mediated by ADH or CYP2E1. The authors should request or generate the recombinant cell line to reproduce the results from HepG2, otherwise no in vitro data should be included.

2. Another major concern is the statistical analysis, the authors did not clearly present in all the figures that which group was used as a control to which other groups were used to compare, and the superscripts a, b, c are very confusing, and the audience did not know which group is truly different from the control group. Please avoid using a, b, c and use lines and asterisk or numbers instead.

3. Serum ALT level in alcohol-feeding mice increased only from 20 to 35, which is hardly to be convincing that this parameter was significantly induced. It seems that only serum AST, serum TG, inflammatory factors like TNFa and IL6, SOD, catalase were significantly altered by EPPH compared to the control group, the partial regulation of the parameters reported in the study should be carefully interpreted because of the inconsistency.

4. The change in lipid content in Figure 8 should be quantified and the result should be reported.

5. Other issues:

Line 67 on page 2, the sentence was not complete, please double check and rephrase it.

Line 76 on page 3, the sentence was also not a complete one.

Materials part 2-1 and 2-2 were repetitive in some description about how to prepare the EPPH, please rephrase the sentences.

Line 181 should provide which actin was used as an internal control.

Fig7, 8 should include scale bar for the images.

6. English proof-reading is needed for the revised version.

Author Response

  1. A major concern is HepG2 is not a suitable cell line for investigating the effect of alcohol in vitro, as HepG2 barely expresses ADH or CYP2E1, as shown in a report (Donohue TM, et al. Recombinant Hep G2 cells that express alcohol dehydrogenase and cytochrome P450 2E1 as a model of ethanol-elicited cytotoxicity. Int J Biochem Cell Biol. 2006 Jan;38(1):92-101. doi: 10.1016/j.biocel.2005.07.010. Epub 2005 Sep 6. PMID: 16181800.) Typically, the recombinant HepG2 derivative cells or primary hepatocytes should be used to that end, otherwise, the results based upon parental HepG2 after alcohol treatment cannot be interpreted as the consequences of alcohol treatment mediated by ADH or CYP2E1. The authors should request or generate the recombinant cell line to reproduce the results from HepG2, otherwise no in vitro data should be included.

We appreciate your comments that may improve the quality of our manuscript. As suggested by the reviewer, in order to express CYP2E1 in HepG2 cells, it is necessary to confirm the expression of CYP2E1 by constructing a CYP2E1 expression vector through protein-encoding region cloning.

In our manuscript, EPPH directly targeted CYP2E1, and this result may lack a mechanism explanation. We are planning a follow-up study of this relevance. In this study, a thesis was written focusing on the improvement of ALD disease through the antioxidant and anti-inflammatory effects of EEPH on various oxidative stress and inflammation induced by the onset of ALD disease.

  1. Another major concern is the statistical analysis, the authors did not clearly present in all the figures that which group was used as a control to which other groups were used to compare, and the superscripts a, b, c are very confusing, and the audience did not know which group is truly different from the control group. Please avoid using a, b, c and use lines and asterisk or numbers instead.

We appreciate your comments that may improve the quality of our manuscript. We had insufficient interpretation of the statistical results of this manuscript. We used ANOVA for the analysis of the data contained in this manuscript. Because we thought it was necessary to compare all groups to analyze the results.

We tried to check the statistical significance between groups through the ANOVA system for the quantitative values of groups in the statistical part of all experimental results.

The alphabets such as a, b, c and d shown in this manuscript were designated as a, b, c and d from the group with the lowest result regardless of the normal group or the induced group.

If the alpha pets between the groups are the same, it was interpreted as 'it is judged that there is no significance between the groups'. And when the alphabets between the groups are different, it was interpreted as 'it is judged that there is significance between the groups'.

In this experiment, it can be confirmed that the treatment of EPPH in HepG2 cells and experimental animal models showed significant results in various ALD-related indicators based on the control group. In our revised manuscript, we added a section that lacked interpretation of the data results.

  1. Serum ALT level in alcohol-feeding mice increased only from 20 to 35, which is hardly to be convincing that this parameter was significantly induced. It seems that only serum AST, serum TG, inflammatory factors like TNFa and IL6, SOD, catalase were significantly altered by EPPH compared to the control group, the partial regulation of the parameters reported in the study should be carefully interpreted because of the inconsistency.

We appreciate your comments that may improve the quality of our manuscript. We agreed with the reviewers' opinions and made efforts to correct the shortcomings in the interpretation of the results. In the process, we confirmed the results of increasing all ALT markers in serum levels of alcohol liver injury animal models through several references.

  1. Astaxanthin alleviated ethanol-induced liver injury by inhibiton of oxidative stress and inflammatory responses via blocking of STAT3 activity
  2. Ethanolic Extract of Acanthopanax koreanum Nakai Alleviates Alcoholic Liver Damage Combined with a High-Fat Diet in C57BL/6J Mice
  3. Dendropanax morbifera Leaf Extracts Improved Alcohol Liver Injury in Association with Changes in the Gut Microbiota of Rats

However, as the reviewer said, there are parts that have not been able to confirm the tendency of concentration-dependent inhibition or increase in ALP, GGT and cholesterol-related areas according to EPPH treatment. Among the animal test results, the parts where it is difficult to accurately determine the tendency were rewritten.

  1. The change in lipid content in Figure 8 should be quantified and the result should be reported.

We appreciate your comments that may improve the quality of our manuscript. We added data quantifying the lipid content in Figure 8.

  1. Other issues:

Line 67 on page 2, the sentence was not complete, please double check and rephrase it. – We appreciate your comments that may improve the quality of our manuscript. We confirmed the reviewer's request and made changes.

Line 76 on page 3, the sentence was also not a complete one. – We appreciate your comments that may improve the quality of our manuscript. We confirmed the reviewer's request and made changes.

Materials part 2-1 and 2-2 were repetitive in some description about how to prepare the EPPH, please rephrase the sentences. - We appreciate your comments that may improve the quality of our manuscript. We confirmed the reviewer's request and made changes.

Line 181 should provide which actin was used as an internal control. – We appreciate your comments that may improve the quality of our manuscript. We used actin as an internal control for western blot results, and additionally submitted a western blot quantitative file.

Fig7, 8 should include scale bar for the images. – We appreciate your comments that may improve the quality of our manuscript. We modified the part requested by the reviewer and attached it.

Round 2

Reviewer 2 Report

1. "Enzymatic porcine placental hydrolyzing enzyme (EPPH)" in line 17 on page 1 should be corrected, as EPPH is hydrolysate not enzyme. Please also double check throughout the manuscript, especially line 41, line 110, which word "hydrolysate" or "hydrolyzate" should be uniformly used. 

2. Although the authors explained in the reply why they used a, b, c on the top of each graph bars, they still need to give interpretation in each legend to each figure to let the audience know what the exact meaning is. 

Author Response

  1. "Enzymatic porcine placental hydrolyzing enzyme (EPPH)" in line 17 on page 1 should be corrected, as EPPH is hydrolysate not enzyme. Please also double check throughout the manuscript, especially line 41, line 110, which word "hydrolysate" or "hydrolyzate" should be uniformly used. : We appreciate your comments that may improve the quality of our manuscript. We confirmed the reviewer's request and made changes.
  2. Although the authors explained in the reply why they used a, b, c on the top of each graph bars, they still need to give interpretation in each legend to each figure to let the audience know what the exact meaning is. : We appreciate your comments that may improve the quality of our manuscript. We confirmed the reviewer's request and made changes.
